# Point-of-Care Diagnosis of Endometrial Cancer Using the Surgical Intelligent Knife (iKnife)—A Prospective Pilot Study of Diagnostic Accuracy

**DOI:** 10.3390/cancers14235892

**Published:** 2022-11-29

**Authors:** Diana Marcus, David L. Phelps, Adele Savage, Julia Balog, Hiromi Kudo, Roberto Dina, Zsolt Bodai, Francesca Rosini, Jacey Ip, Ala Amgheib, Julia Abda, Eftychios Manoli, James McKenzie, Joseph Yazbek, Zoltan Takats, Sadaf Ghaem-Maghami

**Affiliations:** 1Department of Surgery and Cancer, Imperial College London, Du Cane Road, London W12 0NN, UK; 2Department of Gynaecological Oncology, University Hospital Southampton, Coxford Road, Southampton SO16 5YA, UK; 3Department of Metabolism, Digestion and Reproduction, Imperial College London, Du Cane Road, London W12 0NN, UK; 4Centre for Pathology, Imperial College London, 4th Floor Clarence Wing, St Mary’s Hospital, London W2 1NY, UK

**Keywords:** endometrial cancer, mass spectrometer, point of care, diagnosis, Pipelle biopsy, rapid diagnosis

## Abstract

**Simple Summary:**

The iKnife is an emerging tool which uses standard electrosurgical methods to generate surgical aerosols that are then interrogated by a mass spectrometer to provide real-time tissue signatures. It has been shown to correctly identify different tissue types including lung, colon and liver. The aim of this study was to ascertain whether the iKnife can correctly identify endometrial cancer from endometrial Pipelle biopsy samples, thereby facilitating point-of-care diagnosis. One hundred and fifty endometrial Pipelle samples were analysed in this study. The iKnife reliably diagnosed endometrial cancer in seconds, with a diagnostic accuracy of 89%, minimising the current delays for women whilst awaiting a histopathological diagnosis. The findings presented in this study can pave the way for new diagnostic pathways.

**Abstract:**

**Introduction:** Delays in the diagnosis and treatment of endometrial cancer negatively impact patient survival. The aim of this study was to establish whether rapid evaporative ionisation mass spectrometry using the iKnife can accurately distinguish between normal and malignant endometrial biopsy tissue samples in real time, enabling point-of-care (POC) diagnoses. **Methods:** Pipelle biopsy samples were obtained from consecutive women needing biopsies for clinical reasons. A Waters G2-XS Xevo Q-Tof mass spectrometer was used in conjunction with a modified handheld diathermy (collectively called the ‘iKnife’). Each tissue sample was processed with diathermy, and the resultant surgical aerosol containing ionic lipid species was then analysed, producing spectra. Principal component analyses and linear discriminant analyses were performed to determine variance in spectral signatures. Leave-one-patient-out cross-validation was used to test the diagnostic accuracy. **Results:** One hundred and fifty patients provided Pipelle biopsy samples (85 normal, 59 malignant, 4 hyperplasia and 2 insufficient), yielding 453 spectra. The iKnife differentiated between normal and malignant endometrial tissues on the basis of differential phospholipid spectra. Cross-validation revealed a diagnostic accuracy of 89% with sensitivity, specificity, positive predictive value and negative predictive value of 85%, 93%, 94% and 85%, respectively. **Conclusions:** This study is the first to use the iKnife to identify cancer in endometrial Pipelle biopsy samples. These results are highly encouraging and suggest that the iKnife could be used in the clinic to provide a POC diagnosis.

## 1. Introduction

Endometrial cancer is the most common gynaecological cancer in the western world, with over 120,000 new cases diagnosed per year in the European Union [1]. Routine management varies, but typically pelvic ultrasound is performed, followed by endometrial biopsy if warranted. Biopsies can be obtained by outpatient endometrial sampling or by hysteroscopy, from either outpatients or inpatients. Histological diagnosis can take up to two weeks. It has been demonstrated that delays in diagnosis are associated with high levels of anxiety and stress for patients [2,3,4,5,6]. The mainstay of treatment for endometrial cancer is hysterectomy, with large international observational studies showing that delays to definitive surgery can negatively impact survival [7,8,9]. To date, point-of-care (POC) diagnosis of endometrial cancer has not been available.

Rapid evaporative ionisation mass spectrometry (REIMS), also known as surgical intelligent knife (iKnife), can rapidly identify human tissues in real time [10]. REIMS is a form of ambient mass spectrometry (MS). Standard electrosurgical diathermy converts tissue into gas-phase ionic species, and the surgical aerosol is then analysed, nearly in real time, by a time-of-flight (ToF) mass spectrometer [11]. The iKnife has been shown to clearly discriminate between tissue types such as lung, liver and colon based on tissue-derived lipidomic profiles [11]. Furthermore, the iKnife has also been shown to accurately discriminate between normal and malignant tissue across various tumour sites including colon, breast, cervical and ovarian tissues [12,13,14,15]. Aberrant lipid metabolism is a key feature of cancer, due to the increased biosynthesis and generation of bioactive signalling molecules and the formation of new cell membranes [16,17]. It is the culmination of these lipid changes in malignant tissue compared to normal tissue that enable the iKnife to discriminate between diseased and healthy tissues.

The aim of this study was to establish if the iKnife could distinguish between normal and malignant endometrial Pipelle biopsy samples based on their lipidomic profiles. Furthermore, multivariate statistical models were used to provide a reference dataset for real-time tissue recognition in the clinic and a rapid POC diagnosis for endometrial cancer.

## 2. Materials and Methods

A single-centre, prospective clinical trial was performed at Imperial College Healthcare NHS Trust (London, UK). STARD guidelines were followed [18].

### 2.1. Population and Samples

Research Pipelle biopsies were obtained from consecutive patients with suspected endometrial cancer attending the rapid access gynaecology clinic or day surgery (undergoing hysteroscopy) between June 2017 and April 2019. All patients provided written, informed consent. Only patients who had a clinical indication for a Pipelle biopsy could participate in the study. Indications for tissue biopsy included an abnormal scan following post-menopausal bleeding or intermenstrual bleeding or an abnormal incidental finding on imaging. Patients were excluded if they lacked capacity, were under 18 years old, were pregnant, did not understand English, or if there were any contra-indications to carrying out a biopsy.

To ensure all patients received gold standard clinical care, the sample obtained for research was from a second pass of the Pipelle (the first pass sent for histopathology analysis). Patients agreeing for a research Pipelle, but from whom no tissue was obtained, were excluded from the study. The demographic details of the patients were collected, in addition to the final histopathological diagnosis.

### 2.2. The iKnife and Sample Processing

Upon biopsy, the samples were either processed fresh or snap-frozen using a CoolRack and stored at −80 °C within 30 minutes of collection for later processing. For REIMS processing of the frozen samples, the tissue was first thawed at room temperature. The clinician performing the tissue processing was blinded to the tissue diagnosis. Prior to sample processing, the mass spectrometer was calibrated as per the manufacturer’s instructions.

The REIMS method, as previously described [10,11], couples a Waters G2-XS Xevo Q-TOF mass spectrometer with a modified handheld diathermy. Figure 1 illustrates the REIMS set up. Prior to tissue diathermy, the samples were first processed with a CO_2_ laser (l =10.6 µm, maximum power of 10 Watts, Omniguide, USA) followed by diathermy using the Covidien ForceTriad™ generator (cut mode, power setting 15–20 Watts). Both the conventional, handheld diathermy iKnife and the CO_2_ laser convert tissue constituents to gas-phase ionic species that are drawn into the MS inlet and Venturi air pump through the REIMS interface for analysis in negative ion mode. A solution of isopropyl alcohol (IPA) and leucine encephalin (LeuEnk) (1 ng/μL) was sprayed towards the REIMS interface at a flow rate of 0.2 mL/min. Leucine encephalin was used as an external lock mass (554.262 m/z). In addition, a known phospholipid (phosphatidic acid) was used as an internal lock mass (699.497 m/z). Multiple tissue sampling points (burns) were performed to create a surgical aerosol. The number of individual burns was dependent on the tissue size (median = 3). Large endometrial biopsy samples produced multiple spectra, whilst small samples produced only a single spectrum. The scan range and time were set to 15–1500 m/z and one second, respectively. Spectral processing was performed within the 600 to 1000 m/z range.

### 2.3. Histological Validation

The final histological diagnosis for all samples was known (from the concurrently processed clinically indicated Pipelle biopsy), including cell type and grade of the tumours. To ensure the cellular composition of the research Pipelle biopsy was representative of the final diagnosis, histopathological validation of the processed samples was required. Pipelle biopsies are blind sampling devices; it is therefore possible that the research Pipelle could miss a malignant endometrial lesion. Thus, the histopathologist confirmed whether the specific biopsy analysed by REIMS contained benign or malignant tissue. All processed tissues were formalin-fixed, paraffin-embedded (FFPE) and then stained with haematoxylin and eosin (H&E). Digital images of each histological sample were acquired using NDP.view2 software (Hamamatsu Photonics, Hamamatsu, Japan). Tissue samples were all reported by a senior histopathologist and reported according to FIGO recommendations, and this acted as the reference standard. The reference standard is the current gold standard and is considered 100% accurate. If it was not possible to make a diagnosis due to thermal artefacts, the sample was excluded from further analyses. The histopathologist was blinded to the clinical information and the result of the iKnife diagnoses.

### 2.4. Statistical Analysis and Classification Models

In-house software developed by Waters Corporation (Offline Model Builder (OMB), version Abstract model builder, AMX) was used initially for all pre-processing of spectra and for model building. OMB software enables background subtraction, lock mass correction, normalisation, peak alignment and binning of spectral peaks to 0.1 Da. Poor-quality spectra with high noise-to-signal ratio were excluded as were any cases with missing data. In addition, files were also imported to MatLab version 2016a (MathWorks), and this enabled the creation of learning curves. No normalisation was required, as it was performed by OMB.

Multivariate models, incorporating histology and corresponding mass spectra, were created based on one spectrum per sample (thus creating a representative spectrum of all the burns in a specific sample). Multivariate analyses were performed to determine the variance in spectral signatures between tissues, for example, normal versus malignant endometrium. This included 2-dimensional (2D) or 3-dimensional (3D) principal component analyses (PCA) and linear discriminant analyses (LDA). PCA is an unsupervised analysis to visually represent tissue class separation. The percentage of variance for each principal component was calculated and recorded. Supervised LDA was also performed, providing graphical 2D or 3D representations of the data.

Loading plots were created to visualise principal components as 2D score plots. Each principal component loading plot illustrates peaks and corresponding intensities that account for the variance within each principal component for any model.

Validation of each PC-LDA model was performed using leave-one-patient-out cross-validation. This method repeatedly leaves out all spectra from an individual patient sample and rebuilds the model. The removed spectra are then tested against the new model to determine the model’s diagnostic accuracy (sensitivity, specificity, positive predictive value and negative predictive value).

A learning curve was created. This is a pictorial representation of the changes in classification rate as more samples are included in the training set of a predictive model. It shows a change in classification performance (*y*-axis) as a function of the number of observations (*x*-axis). Classification accuracy should improve as more observations are included, up to a finite number, at which point the classification rate plateaus. It is useful in a context such as this to estimate whether maximum diagnostic performance is reached, or if more samples are needed.

### 2.5. Univariate Analysis and Lipid Identification

Data matrices were interrogated by MatLab; the intensity values and median fold change of binned m/z values were analysed using the Kruskal–Wallis test to identify the discriminatory m/z values responsible for separation in the models. *p*-values were reported for each m/z value. To minimise false discovery in the univariate analyses, the *p*-values were corrected to q-values using the Benjamini–Hochberg–Yekutieli method (false discovery rate, α = 0.05).

Loading plot and univariate analysis m/z values, that were abundant and strongly contributed to class separation in the PC-LDA models, were selected for further tandem mass spectrometry (MS/MS) analyses. Methods for MS/MS were similar to those previously described, but argon was used as the collision gas, and the collision energy was set to 30 Watts but varied according to the sample. For the MS/MS experiments, the CO_2_ laser was used for tissue sampling over 2–7 seconds per burn in order to achieve sufficient aerosolised gas for analysis for a specific m/z value. A full scan was performed on each tissue sample prior to MS/MS to confirm the lock mass and that peaks of interest were present.

The spectra from MS/MS negative-ion-mode were averaged and presented to the LipidMaps online tool. LipidMaps (Lipid Metabolite and Pathways Strategy Lipidomic Gateway), is a free online resource sponsored by the Wellcome Trust (20). The following search parameters were used within LipidMaps: [M-H]- and [M+Cl]-, intensity threshold 5, ion mass tolerance +/− 0.1 m/z, any head group. LipidMaps suggested possible lipids and chain configurations for each m/z value. Only even fatty chain compounds were included. The main daughter ion was used to narrow down the exact lipid match.

## 3. Results

### 3.1. Tissue Characteristics and Processing

One hundred and fifty Pipelle samples were obtained. No adverse events were reported. Four patients had endometrial hyperplasia, and two samples were insufficient on final histology due to atrophy; the latter patients were excluded. The remaining 144 samples were classified as either cancer or normal (Table 1). The patients with cancer tended to be older (median age 64 vs. 54 years), with a higher proportion of patients presenting with postmenopausal bleeding (PMB). Most cancer samples were endometrioid type 1 subtype (83%). Of the patients with cancer, 58% had stage 1A disease, 20% stage 1B, and the remainder had more advanced disease.

Due to large variations in the quality of the tissues, a simple quality score was determined for each sample, i.e., poor, moderate and good, which were recorded contemporaneously. Poor samples were defined by very bloody samples with minimal amounts of solid tissue (less than 0.5 cm for any of the diameters). Good samples were defined by large amounts of solid tissue (more than 1 cm for any diameter). Moderate samples had solid elements of 0.5–1 cm.

All 150 Pipelle samples were processed with REIMS. Five samples were predominately liquid medium, and no lipid spectra were obtained with the monopolar diathermy or CO_2_ laser. Furthermore, four samples were too small to be processed with monopolar diathermy and, therefore, were processed with the CO_2_ laser only. The remaining 141 samples were processed with the monopolar diathermy iKnife, producing 453 spectra. Of the 150 samples, 106 samples were also processed with the CO_2_ laser, yielding 106 spectra. Three samples (all benign), processed with REIMS, were excluded from the analyses as no spectra passed the quality control checks (see Appendix A
Table A1). No laser spectra were removed due to poor quality, compared to 52 monopolar diathermy spectra.

The reference standard was the final histological diagnosis based on the H&E slide following tissue sampling with the iKnife; this took place 4–12 weeks post tissue sampling. The H&E slides are very robust, and delays in reporting made no difference for the final histological diagnosis. The histological quality control excluded a large proportion of samples (35% for the iKnife and 39% CO_2_ laser). This was largely due to thermal artefacts caused by the diathermy/CO_2_ laser. It was essential to be robust with the exclusion criteria at this step to ensure that the resultant reference dataset contained only spectra that were truly derived from confirmed histology. Some cancer samples did not have any recognisable cancer tissue remaining during post-processing histopathological analysis, either because the cancer within the sample was burned, or because the Pipelle biopsy sample did not include the intra-uterine malignancy.

### 3.2. Statistical Modelling with the iKnife—Normal Tissue versus Cancer

Only samples which were histologically validated as either normal or endometrial cancer were included in the models. The single validated hyperplasia sample was excluded. Three models were created—the first sampled the tissue using the conventional diathermy iKnife (Figure 2A,B), the second using the CO_2_ laser iKnife (Figure 2C,D) and a third comparing normal tissue to type 1 endometrial cancer samples only (Figure 3A,B). Each data point in principal component analysis incorporates data from all spectra obtained from a single tissue sample. The red points represent cancer samples, and the blue ones, normal tissue. Whilst no apparent separation was seen between tissue classes in PCA, there was separation in PC/LDA based on the high diagnostic accuracies seen on cross validation. For the first model with the diathermy iKnife, the sensitivity, specificity, positive predictive value (PPV) and negative predictive values (NPV) were 81%, 91%, 90% and 84%, respectively. Overall, 86% of patient samples were correctly identified using the cross-validation. The sensitivity, specificity, PPV and NPV of the second model with the CO_2_ laser iKnife were 85%, 93%, 94% and 85%, respectively. Overall, 89% of the patient samples were correctly identified.

In the third model, normal and type 1 cancer samples processed with the diathermy iKnife were used. As before, in the Figure, each dot on the PCA plots represents the average spectra for a single sample. Red points represent type 1 cancer, and blue points represent normal tissue. The overall sensitivity, specificity, PPV and NPV were 79%, 96%, 93% and 86%, respectively. The overall accuracy was 89%.

### 3.3. Learning Curve

In this learning curve (Figure 4), the algorithm started with five observations per class and incremented by one sample per class each time, until reaching 32 samples per class (maximum size of training set). For each training set size, all of the observations were randomly partitioned 20 times (‘resampled iterations’) in order to estimate the classification performance using different observations for training and testing. The mean accuracy, sensitivity and specificity were determined for each iteration and plotted on a learning curve (Figure 4). The pink boundaries represent ± 1 standard deviation of the accuracy. The ‘curve' shows a gradual increase in diagnostic accuracy from 70 ± 10% to approximately 80 ± 10% when increasing the sample size from 5 to 32. It can also be seen that the curve did not plateau, suggesting that an increase in sample size may improve the diagnostic accuracy.

### 3.4. Loading Plot and Univariate Analysis

The loading plots for PC1 of the CO_2_ laser model (cancer versus normal) is depicted below (Figure 5). These only provide an overall visual assessment of the significant m/z values, and the Kruskal–Wallis test was also performed to confirm these results. Interestingly, very few m/z values >800 m/z accounted for the variance in PC1.

Using Matlab, the Kruskal–Wallis analysis was performed to confirm the identify of specific m/z values responsible for the separation of cancer and normal samples on the PC/LDA model. A -log2fold change of 1, in addition to a q value of 0.001, was used as a threshold to reduce the number of significant values.

Table 2 shows some of the key m/z values which were abundant and/or responsible for the separation of classes and lipids involved, identified using tandem MS. The m/z peaks were presented to the database in negative ion mode. LipidMaps proposes deprotonated ions [M-H]- and chlorinated ions [M+Cl]- as well as deaminated ions [M-NH3]-. The lipid groups that were found to be upregulated in endometrial cancer included ceramides and the phospholipids phosphatidic acid (PA), phosphatidylethanolamines (PE) and phosphatidylserines (PS). Tandem MS was not possible for all significant lipids responsible for the separation of classes in the PCA model and loading plots, as the signal was too noisy.

The fragmentation pattern of each ion using tandem MS was interrogated using LipidMaps. A typical MS spectrum of m/z 722.54 is shown in Figure 6. The mother ion can be seen, approximately one-third of the height of the biggest daughter ion (303.24), along with the other daughter ions 259.25, 418.29, 436.30. Using LipidMaps, we confirmed the lipid as a PE (P -16:0/20:4).

## 4. Discussion

### 4.1. Key Findings

This prospective diagnostic study is the first to report the use of REIMS for the real-time diagnosis of endometrial cancer. REIMS provides rapid tissue classification and a result within 1.8 seconds [13] as opposed to 1 to 2 weeks using standard histology, thereby facilitating point-of-care diagnosis. The overall diagnostic performance of the models is encouraging, with a sensitivity as high as 89%. The learning models (Figure 5) suggest that adding more samples to the spectral library will further increase the diagnostic accuracy of the models.

The CO_2_ laser was a better diagnostic tool than the iKnife (overall accuracy 89% vs. 86%). As described by Genangeli et al, the CO_2_ laser provides cleaner cuts, yielding higher quality spectra and less background noise [19]. Unlike the iKnife, which had 15% (70/453) of spectra excluded due to poor quality, the CO_2_ laser had no spectra excluded. The CO_2_ laser energy is rapidly absorbed by water and burns to a depth of approximately 20 µm depending on the tissue type, leaving ample tissue for histological validation [20]. More subtle differences in the phospholipid spectra seen when using the CO_2_ laser may be the reason for the superior diagnostic accuracy of the CO_2_ laser.

In addition to its diagnostic role, the iKnife (both CO_2_ and diathermy) can provide information about the phospholipid constitution of cancer versus normal cells. An important feature of malignancy is uncontrolled cell division, necessitating an increase in cell membrane lipid synthesis. This is reflected by an increase in phospholipid biosynthesis pathways and an alteration in the cellular phospholipid make-up of cancer cells. The iKnife interrogates these changes to distinguish between tissue classes. Multiple studies have suggested that glycerophospholipids can be used as biomarkers for cancer [21,22,23,24]. The loadings plots and univariate analyses in the present study suggested multiple m/z values had differential abundance and intensity in tissue samples, which contributed to class separation. This suggests that the entire lipid signature may be important for class separation rather than single m/z values. It is therefore unlikely that individual lipids could be used as diagnostic markers for endometrial cancer; more likely, a combination of lipids can be used as part of a multivariate prediction model to diagnose endometrial cancer.

### 4.2. Implications and Comparison with Other Studies

At present, both primary and secondary prevention in endometrial cancer is lacking. A large meta-analysis has shown that early detection in women presenting with PMB will capture approximately 90% of endometrial cancer cases [25,26]. Furthermore, those with significant risk factors such as morbid obesity may benefit from risk-reducing interventions [27]. Whilst screening pathways are not fully established, early detection remains possible. Molecular biomarkers such as Ca125 and HE4 have been evaluated with some success but have not been integrated into the clinical practice [28]. Emerging technologies such as REIMS and vibrational spectroscopy may also aid an earlier and more reliable detection.

A recent study by Paraskevaidi et al., using vibrational spectroscopy to look for endometrial cancer biomarkers in blood samples, found a high diagnostic accuracy (71–100%) [29]. The study also found an increase in lipid-related peaks in the blood of women with endometrial cancer. The biggest disadvantage of this study was the authors’ inability to delve deeper and correlate the spectral peaks with the biomolecular and cellular changes underpinning carcinogenesis. Importantly, the group failed to correct for multiple testing and maintained a *p*-value of 0.05, despite testing hundreds of peaks with a high likelihood of a false positive result.

Despite multiple studies showing lipidomic alterations in cancer, few studies have specifically addressed lipid metabolism in endometrial cancer. The lipid assignments for many of the significant m/z values in this study indicated predominately PAs and PEs. This is in keeping with other studies which found higher levels of PAs and PEs in breast, ovarian and bowel cancer [11,12,13,15]. In this study, PE (16:0/18:1), PE (16:0/20:4), PE (18:0/20:4) and PE (20:0/18:2) were all more common in endometrial cancer tissue, when compared with normal tissue. Altadill’s study of 56 endometrial tissue samples found significant dysregulation in lipid metabolism too, with a number of PCs, PEs, PIs and PAs upregulated in endometrial malignant tissue, including PE (16:1/P-18:1). One limitation of that study is the lack of clarity around how the tissue samples were obtained. Other studies have reported profound changes in sphingolipid metabolism in endometrial cancer [30], though this was not demonstrated in the present study. Further cell line work will be needed to unravel the exact molecular pathways involved in endometrial cancer pathogenesis based on the observed differences in phospholipid composition.

### 4.3. Strengths and Limitations

REIMS is a rapidly evolving technology, and this is the first study using it to detect endometrial cancer from Pipelle biopsy samples. The technology performed extremely well, with a diagnostic accuracy of up to 89%. The overall performance of the diathermy iKnife was higher when type 2 samples were excluded. This is unsurprising, as type 1 cancers are more homogenous in nature, with a similar lipidomic profile, compared to type 2 cancers, which present diverse profiles. Type 2 cancers include rare and mixed tumours, and therefore building a robust model for these cancers is more challenging, with fewer individual subtypes within the reference dataset.

The sensitivity and specificity of the best models were sufficiently high to be used in clinical practice, following prospective validation. The result from REIMS is virtually instantaneous and can therefore provide a POC diagnosis for patients attending the clinic. Treatment delays have been shown to be negatively associated with survival [9]. Given the high PPV (90–94%), women who test positive are very likely to have endometrial cancer and could be subjected to blood tests and pre-operative imaging that same day, while awaiting final histopathological confirmation, minimising any treatment delays.

One of the limitations of this study is the limited number of Pipelle samples, with a significant number of them being very small and of poor quality. In a study of over 200 women undergoing Pipelle biopsies, 23% of the samples failed, predominately due to insufficient tissue [31]. In this study, research Pipelle samples were acquired only after obtaining adequate tissue (first pass) for standard histology; thus, the samples were often very small. Indeed, the acquisition of the Pipelle samples was challenging. Patients attending the clinic were anxious and not always happy to participate in the research, particularly in view of the painful nature of the procedure needed to obtain the samples. Those patients who agreed to participate in the research did not always need a Pipelle biopsy, and for those who needed a Pipelle sample, there was often insufficient tissue for research. Additionally, the consistency of the research Pipelle samples was heterogenous (completely liquid to solid tissue), and this posed technical challenges for REIMS and increased the complexity of model classification. For tissues which are very different, large reference datasets are even more important to ensure an adequate number of reference data for varied tissue types. Furthermore, natural variations in normal and cancer tissue introduce complexity. For example, benign samples include inactive endometrium, normal secretory endometrium and benign polyps. Molecular profiling has shown differences in endometrial tissue depending on the timing of the cycle, and also the presence of polyps or fibroids [32]; these changes may result in downstream phenotypic changes which are recognised by REIMS.

Another limitation of REIMS is its destructive nature. A high proportion of samples were excluded, as the thermal damage caused by REIMS made histological validation impossible. When this study was first conceived, it was impossible to predict how many samples would be needed for the model to recognise differences in the lipidomic profiles between cancer and normal tissue (or, indeed, if the analysis was even possible). As seen in the learning curve, further improvements in diagnostic accuracy may still be reached with the inclusion of more samples in the model. Further development of the reference dataset will be essential for prospective validation. Care needs to be taken to ensure that sample conditions are as uniform as possible and continue to adopt standard operating procedures for the acquisition, processing, utilisation and storage of tissue to minimise confounding variables [33]. It is well recognised that endometrial pathophysiology is highly complex and that there are multiple sources of biological variability that can affect tissue composition, including body mass index, ethnicity, medications and timing of the biopsy in relation to the menstrual cycle [33].

## 5. Conclusions

The results of this study suggest that REIMS has the potential to expedite the patient pathway, providing point-of-care diagnosis for women with suspected endometrial cancer. It accurately distinguishes normal from malignant endometrial tissue based on differences in their lipidomic profiles. Further larger studies are needed to validate this technique and improve its diagnostic performance.

## Figures and Tables

**Figure 1 cancers-14-05892-f001:**
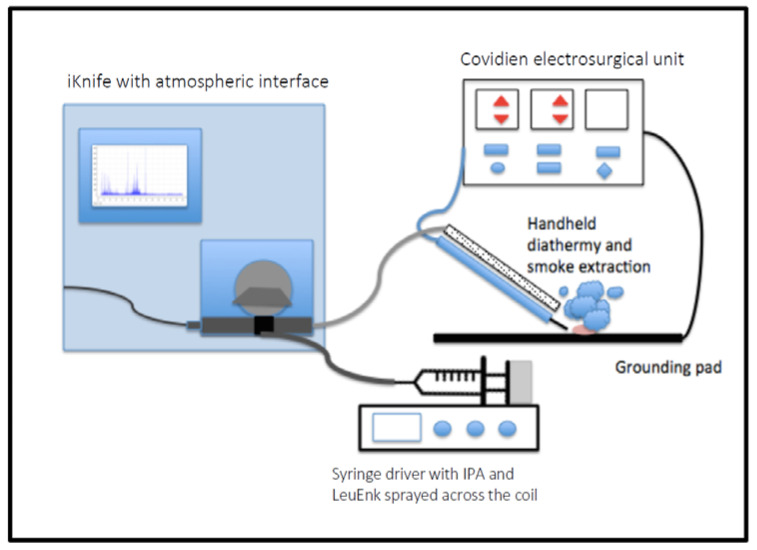
Diagram of the iKnife set-up. Electrical current produced by the generator was applied to the tissue using a handheld diathermy. The charged aerosol produced was extracted through the handpiece and drawn into the iKnife interface and analysed by the G2-XS Xevo Q-ToF. IPA containing LeuEnk (leucine encephalin) was sprayed across a heated coil.

**Figure 2 cancers-14-05892-f002:**
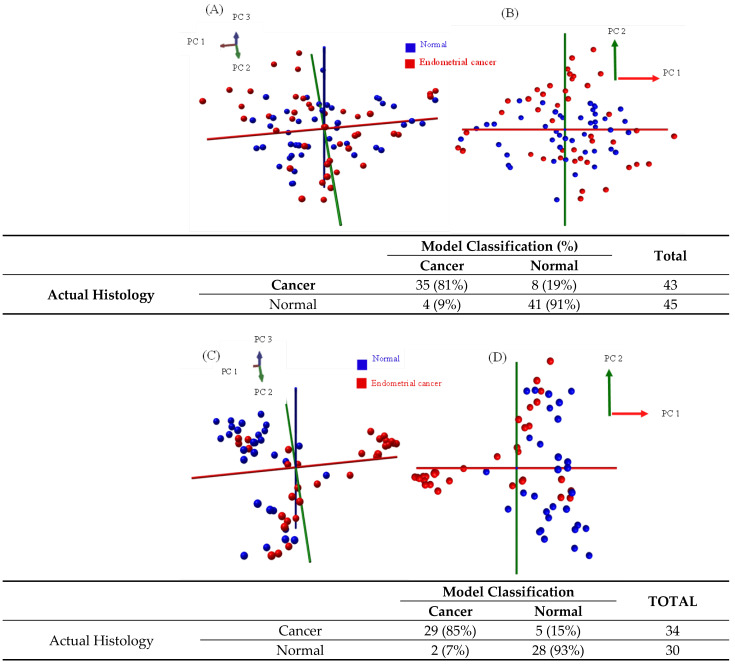
Multivariate analysis of normal versus cancer Pipelle samples after histological validation, processed with the standard diathermy iKnife (**A**,**B**) and CO_2_ laser iKnife (**C**,**D**). Red data points represent endometrial cancer samples and blue points represent normal samples. Each point represents the average spectrum for a single tissue sample. Images **A** and **C** are 3D PCA plots, and images **B** and **D** are 2D PCA plots. Principal component one accounted for 34.1% of the variance in the diathermy iKnife model and 42.7% in the CO_2_ laser model. The confusion tables are shown alongside the figures. The tables show the accuracy of the model classification compared to histopathological gold standard.

**Figure 3 cancers-14-05892-f003:**
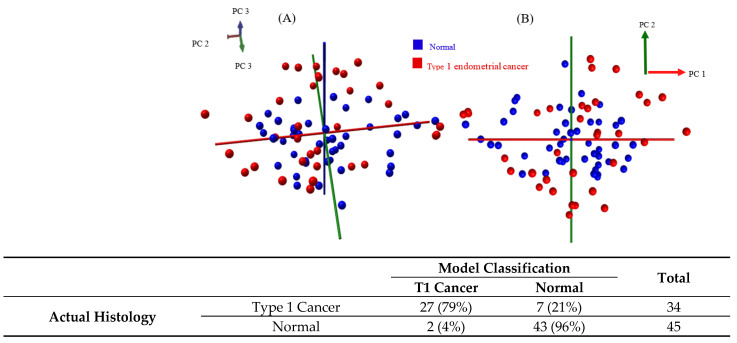
Multivariate analysis of normal versus type 1 cancer Pipelle samples after histological validation, processed with the standard diathermy iKnife (**A**,**B**). Red data points represent endometrial type 1 cancer samples, and blue points represent normal samples. Each point represents the average spectrum for a single tissue sample. Image **A** is a 3D PCA plot, and image **B** is a 2D PCA plot. Principal component one accounted for 35.8% of the variance in the model, and principal component two accounted for 16.7% of the variance. The confusion table is shown below. The tables show the accuracy of the model classification compared to histopathological gold standard. T1 = type one.

**Figure 4 cancers-14-05892-f004:**
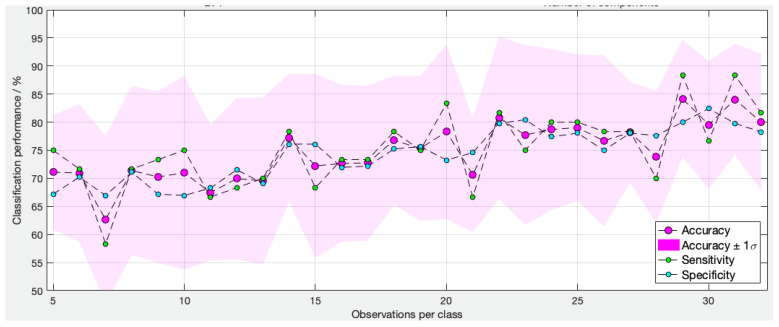
Learning curve of type 1 cancer versus normal (no hyperplasia) diathermy iKnife model based on 35 normal and 35 cancer samples. The algorithm started with five normal and five cancer samples and increased by increments of one. The overall diagnostic accuracy (pink dots) increased from 70 ± 10% to approximately 80 ± 10% when the sample size increased from 5 to 32.

**Figure 5 cancers-14-05892-f005:**
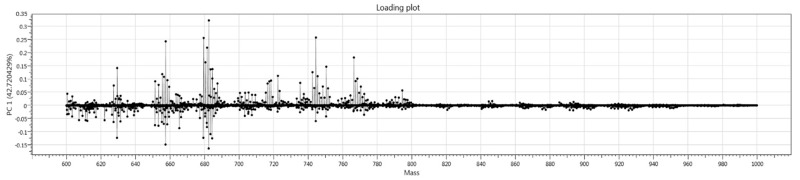
Loading plot of Principal component 1 for the CO_2_ laser model cancer versus normal, no hyperplasia.

**Figure 6 cancers-14-05892-f006:**
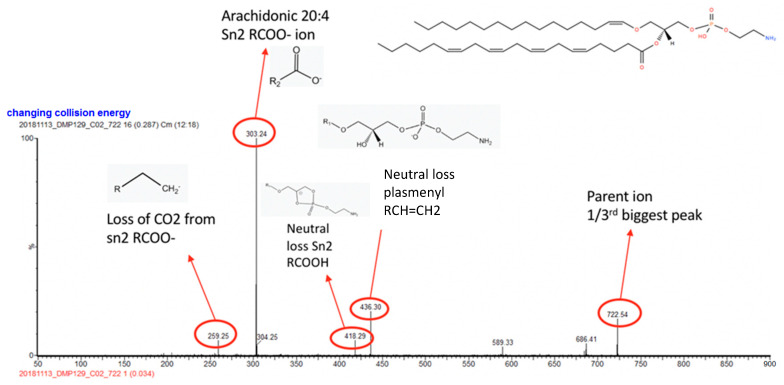
Tandem MS spectra of two m/z values: 722.54. The parent and key daughter ion peaks are annotated along with a chemical illustration of the confirmed phospholipid.

**Table 1 cancers-14-05892-t001:** Summary table of the patient demographics from which the Pipelle biopsies were obtained, along with the samples’ quality. Age (mean and range). PMB = post-menopausal bleeding, HMB = heavy menstrual bleeding, IMB = intermenstrual bleeding, abdo = abdominal, G = grade.

Demographics	Normal (*n* = 85)	Cancer (*n* = 59)
**Age**	54 (22–80yo)	64 (35–85yo)
**Presentation:** **PMB** **HMB** **IMB** **Abdo pain** **Other**	36 (42%) 26 (31%) 10 (12%) 11 (13%) 2 (2%)	49 (83%) 3 (5%) 0 (0%) 5 (8%) 2 (3%)
**Type of Cancer**	N/a	Endometrioid Low grade: 30 (51%) High grade (G2/3) 19 (32%) Serous 7 (12%) Carcinosarcoma 2 (3%) Clear cell 1 (2%)
**Quality Pipelle:** **Small/Bloody** **Moderate** **Good/Polyp**	40 (47%) 22 (26%) 23 (27%)	14 (24%) 9 (15%) 36 (61%)

**Table 2 cancers-14-05892-t002:** Top 10 most abundant m/z values seen in normal and malignant endometrial Pipelle biopsies, along with the lipid assignments following MS/MS. Double carbon bonds not reported. * = MS/MS not possible due to low signal intensity, so only tentative assignment. PI = phosphatidyl-inositol.

Model	m/z Value	*p* Value	Potential Lipid
Cancer vs. Normal CO_2_ laser	600.55	0.00000137	Cer (36:1;O2) *
	671.45	0.000000345	PA (34:2) *
	679.55	0.00000265	PG (O-16:0/14:0) *
	699.55	0.000000340	PA (36:2) *
	722.54	0.0000600	PE (16:0/20:4)
	738.51	0.0259	PE (P-16:0/20:4) (12OH[S])
	750.57	0.00680	PE (18:0/20:4)
	766.56	0.0282	PE (18:0/20:4)
	770.45	0.00000631	PE (20:0/18:2)
	786.55	0.0000000559	PS (36:2) *
	794.57	0.00426	PC (O-18:0/20:4)
	794.65	0.00363	PE (20:4)
	864.57 *	0.000177	PI-Cer (d18:0/22:0)

## Data Availability

The data presented in this study are available on request from the corresponding author. The data are not publicly available due to restrictions on ethics approval.

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
