# Peer review of "Point-of-Care Diagnosis of Endometrial Cancer Using the Surgical Intelligent Knife (iKnife)—A Prospective Pilot Study of Diagnostic Accuracy"

_cancers, 2022, doi:10.3390/cancers14235892_

Round 1

Reviewer 1 Report

General comments.

This manuscript presents the results of a prospective pilot study of the diagnostic accuracy of the surgical intelligent knife for endometrial cancer.

Basically, this is a well-written paper. 

I would recommend it for acceptance after the minor points listed below are addressed.

Specific comments.

Line 219

If possible, please provide a breakdown of FIGO stages for the 59 endometrial cancer cases.

Line 221

In Table 1, please indicate the criteria for pipelle quality small, moderate and good.

Line363

Was there any relationship between the power of the Covidien ForceTriad™ generator (15-20 watts) and the quality of the spectra?

Line 420

What is the minimum size of a sample that is sufficient for research?

Author Response

Many thanks for your kind review and consideration.

Line 219, added breakdown of stage within text

Line 221 added sentence explaining classification of quality of pipelle

Line 363 , no formal analysis so not mentioned in text. High power settings caused excessive charring and made histopathological validation impossible and subsequent exclusion of tissue sample from analysis. Lower settings did not aerosolise tissue sufficient to obtain a spectra that passed quality control. Thus, between 12-20 was optimum for obtaining spectra and enabling histopathological validation.

420 - This was the first study of its kind testing mass spec on endometrial pipelles. So there was no specific sample size. Limitation was due to resources (time to collect samples). Later, the learning curve was created to ascertain whether reached plateau of accuracy. Comparing other lab groups investigated the application of iknife in other cancers, they obtained 50-200 samples. 

Reviewer 2 Report

I read with great interest the Manuscript titled "Point of care diagnosis of endometrial cancer using the surgical intelligent knife (iKnife); a prospective pilot study of diagnostic accuracy".

The subject is interesting given the possible influence on the diagnostic and therapeutic choices that the use of iKnife for the treatment of endometrial cancer could promote.

In my honest opinion, the topic is interesting enough to attract the readers’ attention. Methodology is accurate and conclusions are supported by the data analysis. Nevertheless, authors should clarify some point and improve the discussion citing relevant and novel key articles about the topic.
- Inclusion/exclusion criteria should be better clarified by extending their description.

-What are the implications of these findings for clinical practice and/or further research? It is important to report the results obtained by the authors in the context of clinical practice and to adequately highlight what contribution this study adds to the literature already existing on the topic and to future study perspectives.

-I suggest also authors to organize better the discussion section following this ideal structure: main findings of the study, strength and limitations of the study, implications and comparison with literature, future directions.

-Discussions can be expanded and improved by citing relevant articles (I suggest authors to read and insert in references the following article PMID: 36141217)

Considering all these point,I think it could be of interest for the  readers of this Journal and, in my opinion, it deserves the priority to be published after minor revisions.

Author Response

Many thanks for your kind consideration and review.

In regards to specific comments:   - Inclusion/exclusion criteria expanded   - Expanded on implications of these findings for clinical practice.   - Introduction of subheadings to discussion section as suggested.   - Addition of relevant references suggested and expanded on discussion.   Many thanks